# Refine Synthesized Action-Conditioned 3D Human Motion with Unsupervised Learned Motion Concepts

## Abstract

As a fundamental aspect of human motion understanding, numerous efforts have been devoted to 3D human motion synthesis and tremendous progress has been made in recent years. Nevertheless, the synthesis of natural and seamless human motions still poses challenges, as it is inevitable for flawed frames to occur within the generated sequences. In light of this, there is a substantial demand for a refinement algorithm, an area that has received limited attention in previous research. In this work, we present motion concepts, which are unsupervisedly learned from a set of real motion sequences, to capture the common and regular patterns in human actions. By leveraging motion concepts, we propose a three-step framework to recognize and refine the flawed frames in an action-conditioned motion sequence hierarchically. Exhaustive experiments conducted on two widely-used benchmarks with five representative motion synthesis approaches, demonstrate that our refinement framework significantly elevates the performance of existing approaches, by improving the realism of synthesized motions while simultaneously enhancing their diversity and multimodality. **Our code will be made publicly available.**

## 1 Introduction

Recent years have seen tremendous progress in modeling human motions, especially in the field of action-conditioned human motion synthesis Guo et al. (2020); Petrovich et al. (2021); Chen et al. (2023); Zhang et al. (2024). Though abundant frameworks have already been proposed for this task, synthesizing natural and seamless motion sequences still remains challenging. One of the most formidable hurdles lies in the occurrence of flawed frames during generation, which may prominently harm the realism of the motion sequence. Figure 1 provides some representative illustrations of this situation. We summarize that these cases typically arise in two aspects: i) **anchor** (key) **frames** that can saliently describe the characteristic of an action and ii) the **transition frames** that serve as a smooth change from one anchor to another.

As it is inevitable for existing synthesis frameworks to produce flawed frames to varying degrees, a refinement network is in substantial demand to pose guidance and constraints on the generated sequences, aiming to rectify the artifacts and thereby improve their quality. However, the domain of motion refinement has received scant attention to our knowledge. Inspired by previous research on motion primitives Kulal et al. (2021; 2022), we present motion concepts in this paper, which are unsupervisedly learned from a set of real motions, to capture the common and regular patterns in anchor frames and transition frames of human actions. By leveraging the benefits of motion concepts, a three-step framework is designed to recognize and refine the flawed anchor frames and transition frames in a given sequence hierarchically.

The motion concepts are presented to learn high-level abstractions of the commonality in human actions. According to the aforementioned two aspects that flawed frames may occur, we learn **anchor and transition concepts** correspondingly. The anchor concepts reveal the leading dynamics of an action, while the transition concepts reflect the core of detailed movement patterns.

Given a set of motion sequences, we unsupervisedly learn motion concepts in the following manner. First, we collect anchor and transition frames from motion sequences. Inspired by Kulal et al. (2021;

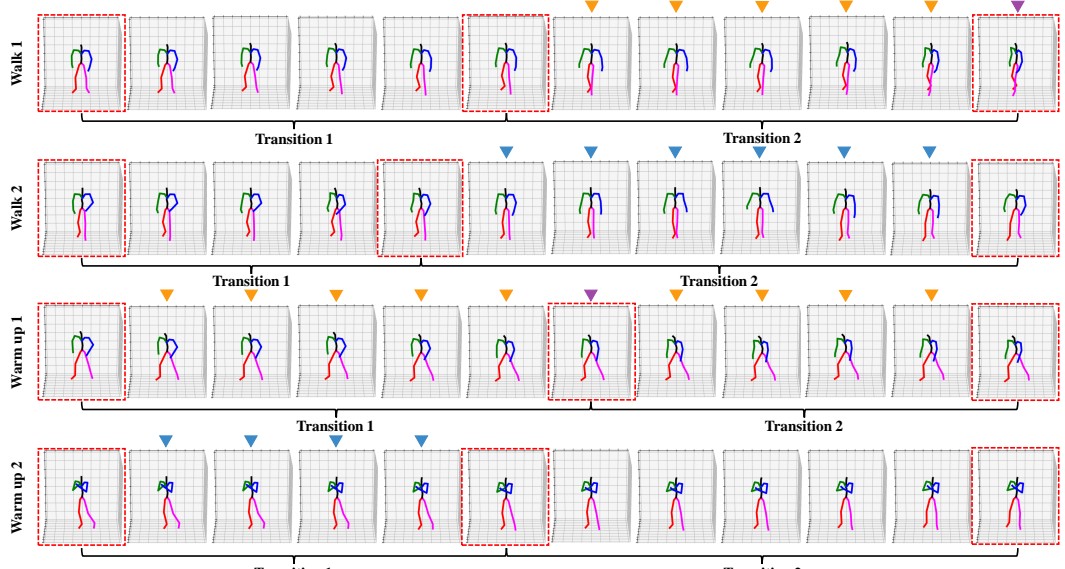

Figure 1: Illustration of synthesized human motion sequence with flawed frames. The frames set out by a dashed red box are anchor frames, while those within two adjacent anchors belong to a transition. The illustrated four motion sequences suffer from flawed frames in either the anchor frames or the transition frames. The frames marked with purple or blue inverted triangles are flawed anchor or transition frames respectively. The frames marked with orange inverted triangles indicate flawed transitions due to flawed anchor frames. For 'Walk 1', the agent should take the right step after taking a left step. But it takes a left step again. For 'Walk 2', the agent attempts to take the right step as shown in 'Transition 2'. But it retracts the right foot first and then gets out. In 'Warm up 1', the agent is doing a lunge leg press exercise. Nonetheless, it fails to press its leg down with flawed anchor frames. For 'Warm up 2', the agent is circling its wrist and ankle. However, we can hardly see the circling movement, indicating the synthesized transition is of low quality.

2022), we consider the whole sequence is composed of consecutive transitions and use splines, a widely used and compact representation of general curves, to fit each transition. Thereby, an entire sequence can be segmented into multiple transitions by finding the best positions of each spline with dynamic programming, and the junction frame between two transitions can be considered as an anchor frame. Then, we explore the common patterns of anchors through a clustering process. All the anchor frames are clustered into several classes, with each class including anchors of a specific pattern. In this way, the cluster center can be properly regarded as an anchor concept. Finally, we model the transition concepts within a distribution learned by a CVAE conditioned on the anchor frames at both ends of a transition.

To improve the artifacts in generated sequences utilizing the learned motion concepts, we design a refinement framework accordingly, consisting of three procedures: i) recognize the anchors in a given motion sequence, ii) refine inappropriate anchors, and iii) refine the transitions. Following this pipeline, we propose three modules, an **anchor recognition module** to locate each anchor of the motion sequence and further identify whether it is flawed, an **anchor refinement module** to synthesize improved anchors to substitute the original anchors with lower quality, and an **transition refinement module** that refines each transition to make the sequence more smooth and harmonize.

To summarize, our contributions are: **1)** We unsupervisedly learn two types of motion concepts to capture the commonality in human motions, namely **anchor** and **transition**. They provide valuable guidance to refine the two kinds of flawed frames inevitably occurring in action-conditioned synthesized human motions. **2)** We propose a three-step framework to refine the flawed frames hierarchically by leveraging our learned motion concepts. **3)** To the best of our knowledge, this work is the first refinement framework for the task of human motion synthesis. Extensive experiments conducted on two widely used benchmarks with five representative baselines demonstrate the superiority of our approach.

## 2    RELATED WORK AND MOTIVATION

**Action-Conditioned 3D Human Motion Synthesis**    Many previous attempts have been made to generate 3D human dynamics from action labels Guo et al. (2020); Petrovich et al. (2021); Tevet et al. (2022); Xu et al. (2023); Chen et al. (2023). Action2Motion Guo et al. (2020) suggests a VAE-based action generation method using Lie algebra as motion representation. ACTOR Petrovich et al. (2021) utilizes a Transformer VAE to generate SMPL human bodies for a given action class. MotionDiffuse Tevet et al. (2022) is the first diffusion-based generative model that generates diverse motions. MLD Chen et al. (2023) performs a diffusion process on the motion latent space which achieves diverse motion generation with low computational overhead. MoGenTS Yuan et al. (2024) generates human motion by jointly modeling spatial body structures and temporal dynamics using a unified framework. While they demonstrate promising results on action-conditioned human motion synthesis, the synthesized motion still inevitably exhibits flawed frames, indicating a non-negligible need for refinement.

**Human Motion Refinement**    Refinement has been adapted to various domains related to human motion such as mocap data acquirement Wang et al. (2016), human pose estimation Luo et al. (2020), and human motion prediction Wei et al. (2023); Wang et al. (2024). For mocap data acqui-sition, Wang et al. Wang et al. (2016) solve the missing marker problem in mocap data through a refinement method based on sparse coding and dictionary learning. In 3D human pose estimation, MEVA Luo et al. (2020) refines the estimated coarse human motion by proposing a residual esti-mation that adds person-specific motion details. For motion prediction, SynSP Wang et al. (2024) utilizes the tension relationship between smoothness and precision to refine the predicted pose se-quence. While significant efforts have been made to refine human motion in multiple scenarios, there is little attention devoted to the task of human motion synthesis. Further, previous motion refinement methods can be hardly applied to human motion synthesis, as they focus on different aspects. Therefore, it is valuable for a refinement approach to the task of motion synthesis.

**Primitive-based Representations for Vision**    The learning of visual concepts within deep net-works has been demonstrated across numerous domains Mao et al. (2019); Tian et al. (2019); Kulal et al. (2022). NS-CL Mao et al. (2019) jointly learns the visual representations and concept repre-sentations, enhancing performance and data efficiency in Visual Question Answering (VQA). Tian *et al.* Tian et al. (2019) propose 3D shape programs to capture low-level geometry and high-level structural priors for 3D shape understanding. PMC Kulal et al. (2022) extends the idea in 3D shape programs and introduces a hierarchical motion representation to capture the structure of 2D human motions to complete various human motion tasks. As parameterized curves Gleicher (1997); Gle-icher & Litwinowicz (1998); Rose III (1999); Kulal et al. (2021); Liu & McMillan (2006); Ghorbel et al. (2015) are widely used as compact and human-interpretable representations for modeling mo-tions, we employ cubic splines to capture the anchor and transition concepts of motion sequences.

## 3    METHODOLOGY

Given a synthesized 3D human motion sequence, $\mathbf{M} = \{P_i | i = 1, ..., N\}$, we are motivated to rectify those flawed frames potentially existing within $\mathbf{M}$ to get a refined motion sequence $\mathbf{M}^* = \{P_i^* | i = 1, ..., N\}$. Here $P_i \in \mathbb{R}^{J \times 3}$ represents the 3D human joints' position at frame $i$, where $J$ is the number of joints in the human skeleton. In this section, we present the unsupervisedly learned motion concepts and the corresponding motion refinement framework.

### 3.1    MOTION CONCEPT

We propose motion concepts to describe two significant essences in human action. One is **anchor**, which indicates the leading dynamics of an action. The other is **transition**, reflecting the core of detailed movement patterns. We learn motion concepts from real human motion sequences in an unsupervised manner.

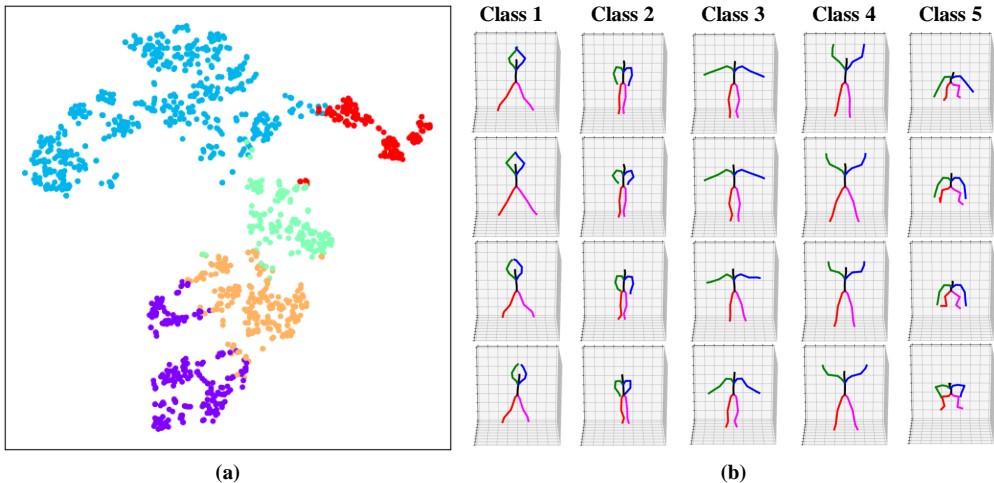

(a)                                    (b)

Figure 2: Visualization of the clustering results of 'Jump'. **a.** Clusters are clearly separated in t-SNE space. **b.** Anchor frames in each cluster show a specific common pattern.

### 3.1.1 DATA COLLECTION

The initial step is to prepare anchor and transition data. Inspired by Kulal et al. (2021; 2022), an action sequence can be seen as consecutive transitions, where the junction frame between two transitions is regarded as an anchor frame.

For a motion sequence $\mathbf{M} = \{P_i | i = 1, ..., N\}$, we firstly segment it into a set of transitions as $\{\{P_i | i = 1, ..., k_1\}, \{P_i | i = k_1, ..., k_2\}, ..., \{P_i | i = k_m, ..., N\}\}$. Following previous research Kulal et al. (2021; 2022) on primitive-based representations which sheds light on this issue, we use spline, a widely used and compact representation of general curves, to fit each transition along each axis for every joint. A spline in 3D space is defined by

$$spline = \begin{bmatrix} X = a_x t^3 + b_x t^2 + c_x t + d_x \\ Y = a_y t^3 + b_y t^2 + c_y t + d_y \\ Z = a_z t^3 + b_z t^2 + c_z t + d_z \end{bmatrix}. \tag{1}$$

It captures the temporal coherence and continuity within the transition using a small and same number of parameters for abstraction Liu & McMillan (2006). Furthermore, we can effectively recover the core movement of human motion through spline interpolation. Then, we use dynamic programming to find the best positions $[k_1, k_2, ..., k_m]$ (*i.e.*, the ones that minimize the overall fitting error) for segmenting the entire sequence into a series of splines. The recurrence relation for the best-fit sequence of splines for the first $n$ frames is given as

$$Error_n = \min_{k<n}[Error_k + fit(P[k:n]) + \lambda], \tag{2}$$

where $Error_k$ is the overall fitting error for the best-fit sequence of splines for the first $k$ frames, and $fit(P[k : n])$ returns the fitting error of a single spline from frame k to n. The parameter $\lambda$ controls the granularity of the fitted splines. In this manner, the motion sequence is processed into a spline sequence, $[s_1, s_2, ..., s_{m+1}]$, of which $s_j = \{a_i^j, b_i^j, c_i^j, d_i^j, T_s^j | i = x, y, z\}$ signifies the transition $\{P_i | i = k_{j-1}, ..., k_j\}$. Here $T_s^j$ denotes the duration of spline $j$. Then, the junction frames, $\{P_{k_1}, P_{k_2}, ..., P_{k_m}\}$, are collected as anchor frames. In our implementation, we default to considering the first and the end frames of the sequence as anchor frames as well.

### 3.1.2 ANCHOR CONCEPT

After obtaining the set of anchor frames $\mathcal{A}$, we can learn anchor concepts, *i.e.* common and regular patterns of anchor frames, with a clustering process. We use the K-means algorithm to cluster the anchors into $n$ classes and the L2 distance to measure the distance between two pieces of anchor data. After clustering, we can properly regard the clustering center of class $i$ as an anchor concept

$A_i$. Leveraging the action label provided in the dataset, the clustering can be performed action by action for better results. We visualize the clustering results of action 'jump' in Figure 2 as an example. By exploring anchor concepts, we can gain a deep understanding of how an action evolves over time, which is crucial for motion refinement.

### 3.1.3 Transition Concept

As a transition fills in the frames between two adjacent anchors, we model the transition concepts within a CVAE transition generator. Given a pair of anchors and a duration $T_s$ as conditions, the generator samples a transition concept represented in the form of spline parameters, describing the core movement patterns to bridge the given two anchors.

To train the transition generator, each piece of collected transition data is utilized by providing the generator with the anchor frames at both ends, along with the duration, as conditions. The entire transition sequence is employed for supervision with the following three loss terms. i) **Reconstruction loss on transition** ($\mathcal{L}_\text{T}$). We use the Mean Per Joint Position Error (MPJPE) loss between the ground-truth transition frames and the restored transition frames according to the spline parameters of the generated motion concept. ii) **Reconstruction loss on the final anchor** ($\mathcal{L}_\text{A}$). To minimize the disparity between the final anchor and better constrain the tendency of the transition, we incorporate an MPJPE loss between the ground-truth final anchor and the restored one. iii) **KL loss** ($\mathcal{L}_\text{KL}$). As in a standard CVAE, we regularize the latent space by encouraging it to be similar to a Gaussian distribution. This is achieved by minimizing the Kullback-Leibler (KL) divergence between the encoder distribution and this target distribution. The total loss is defined as the summation of the above three terms:

$$\mathcal{L} = \mathcal{L}_\text{T} + \lambda_\text{A}\mathcal{L}_\text{A} + \lambda_\text{KL}\mathcal{L}_\text{KL}. \tag{3}$$

We set $\lambda_\text{A}$ to 0.5 and $\lambda_\text{KL}$ to 0.0001 for better performance.

### 3.2 Motion Refinement

Leveraging the learned motion concepts, we design a motion refinement framework to recognize and refine the flawed frames in a given sequence. The framework consists of three components, namely anchor recognition module, anchor refinement module, and transition refinement module.

### 3.2.1 Anchor Recognition Module

Anchor recognition module recognizes the anchors within the sequence as well as identifies whether an anchor is flawed. As shown in Figure 3-a-1, it takes the given motion sequence $\mathbf{M} = \{P_i | i = 1, ..., N\}$ as input, and outputs two vectors of length $N$ as recognition results. The first vector indicates whether frame $i$ is an anchor frame. And for the frames recognized as anchors, we further assess their quality as an anchor and record their scores in the second vector. Those anchors with low scores are considered flawed anchors, indicating a need for refinement.

We use a motion encoder based on the transformer architecture to obtain the representation of each frame within the sequence. The structure of the encoder is illustrated in Figure 3-a-2. First, the motion sequence is linearly embedded and then added by the positional encodings in the form of sinusoidal functions. Second, the features are sent to the transformer encoder to obtain the representation of the sequence $R$, which is then fed to the transformer decoder. Third, we utilize the positional encodings to retrieve the representation of frame $i$ as $R_i$ from the transformer decoder. Once each frame of the motion sequence has been embedded into a deep feature, we can use an MLP classifier to identify the anchor confidence. The motion encoder and classifier are supervised by Binary Cross Entropy Loss as

$$\mathcal{L} = -\frac{1}{N}\sum_{i=1}^{N}(y_i \log(p_i) + (1 - y_i)\log(1 - p_i)), \tag{4}$$

where $p_i$ denotes the output anchor probability of frame $i$ and $y_i$ denotes the ground-truth anchor label of frame $i$.

In terms of assessing the quality, we score the feature of recognized anchors by a Gaussian Mixture Model (GMM) trained with Maximum Likelihood Estimation (MLE) with the collected set of anchor

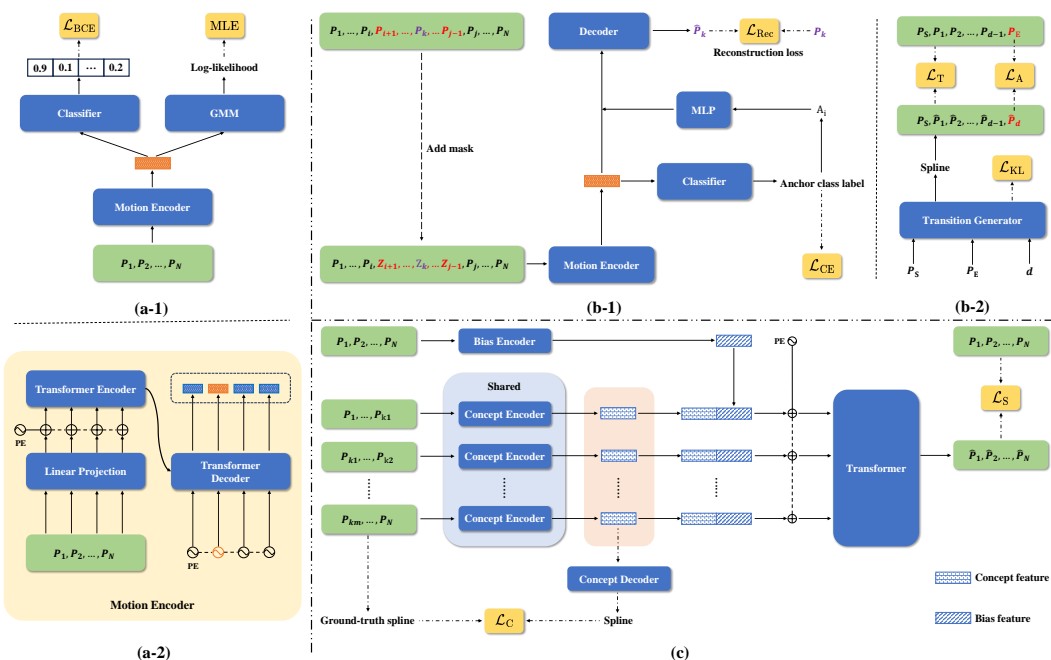

Figure 3: Overview of the proposed motion refinement framework. The green rectangular blocks represent the motion sequences. The blue rectangular blocks denote the neural networks. The yellow rectangular blocks signify the loss functions used for training these networks. The small textured rectangles indicate features. Solid lines depict processes that are followed during both training and testing, whereas dotted lines represent processes that occur only during training. **a.** The anchor recognition module. As shown in Figure a-1, we adopt a classifier to recognize the positions of anchors and a GMM to identify whether an anchor is flawed. Figure a-2 shows the structure of the motion encoder, which obtains the representation of each frame based on the transformer architecture with positional encoding as a query. **b.** The anchor refinement module. In Figure b-1, after selecting the anchor feature from the motion encoder, a classifier is used to identify the anchor class and an MLP decoder is to generate the refined anchor. Figure b-2 explains how we use the transition generator to obtain a transition represented as a spline by giving a pair of anchors and a duration. This helps us to rebuild the neighboring transitions of the flawed anchor. **c.** The transition refinement module, which incorporates the concept feature and the bias feature of a transition to produce the refined motion sequence.

frames. It receives the feature of the recognized anchor and outputs a log-likelihood score indicating the anchor's consistency with the learned multi-dimensional Gaussian distribution. The resulting log-likelihood score serves as the quality score.

### 3.2.2 ANCHOR REFINEMENT MODULE

As anchors dominate the progression of an action, flawed anchor frames often result in substantial deviations from the overall dynamics of the action. Therefore, our anchor refinement module aims to synthesize improved anchors to substitute the flawed ones, instead of making modifications to themselves.

Its detailed structure is in Figure 3-b-1. Given a motion sequence, we start by masking each flawed anchor together with its neighboring transitions to get rid of their impacts in the sequence, and then send the masked sequence to the motion encoder to acquire the feature at the anchor position. With this anchor feature, we can use a classifier to identify which class this anchor belongs to and obtain the anchor concept of this class. Subsequently, by concatenating the anchor feature with the embedding of the anchor concept, we can obtain a new anchor as a substitution for the flawed one through an MLP decoder. As the synthesized anchor might differ significantly from the original one, we recalculate the transition concept of neighboring transitions for the continuity and smoothness of the entire motion sequence with the transition generator. The transition concept can be restored to transition frames with the spline parameters.

To train the anchor refinement module, we randomly select anchor frames in the real motion sequences and add masks in the same way as mentioned above by assuming the chosen anchor is a flawed one. The ground-truth anchor concept is used as input for the decoder. We use Cross Enropy Loss to supervise the training of the classifier and use MPJPE as the reconstruction loss to supervise the generation of the masked anchor frames.

### 3.2.3 Transition Refinement Module

The transition refinement module, as depicted in Figure 3-c, refines the flawed frames in consecutive transitions to enhance the coherence and realism of a motion sequence. Initially, we employ a concept encoder to explore the feature of the transition concept for each piece of transition in the given sequence. The concept feature is responsible for constraining the core movements in a transition process to be smooth and natural. Considering that the distinctive behavioral preferences of different individuals will induce individual variations when they perform the same actions, we further incorporate a bias encoder to capture a bias feature throughout the entire motion sequence. The bias feature complements the concept feature by incorporating individual variations, and the concatenation of both features fully represents a transition. Finally, the summation of the feature and positional encoding is sent to a transformer to model their correlations and produce the refined motion sequence $\mathbf{M}^*$.

The training of the transition refinement module contains two loss terms. i) **MPJPE loss on the concept feature** ($\mathcal{L}_\mathrm{C}$). We supervise the concept feature by the fitted spline. First, the concept feature is decoded into the parameters of a spline with an MLP. Then, we calculate the MPJPE loss between the restored motion sequences from the ground-truth spline and the decoded spline. By minimizing the MPJPE loss, the concept feature can well represent a transition concept. ii) **MPJPE loss on the entire sequence** ($\mathcal{L}_\mathrm{S}$). As it is difficult to isolate behavioral bias from a motion sequence, we refrain from directly supervising the bias feature. Instead, we utilize the MPJPE loss between the ground-truth motion sequence and the refined motion sequence, and backward the gradient to the whole network including the bias encoder. The total loss is the weighted summation of the above two loss terms:

$$\mathcal{L} = \mathcal{L}_\mathrm{S} + \lambda_\mathrm{C}\mathcal{L}_\mathrm{C}, \tag{5}$$

here we set $\lambda_\mathrm{C}$ to 1 to balance these two terms.

## 4 Experiments

### 4.1 Datasets, Evaluation Metrics and Baselines

We conduct our experiments on two widely-used datasets for action-conditioned motion synthesis, namely HumanAct12 Guo et al. (2020) and UESTC Ji et al. (2018). These datasets encompass a significant number of motions with diverse styles. We comprehensively evaluate the quality of motion sequences after refinement with five metrics: Frechet Inception Distance (FID), Recognition Accuracy, Diversity, Multimodality, and Action-FID (A-FID). A-FID evaluates the FID of a specific action. We introduce this metric to provide a more precise assessment of different concepts' effectiveness compared to evaluating the FID of the entire dataset encompassing all actions. We perform motion refinement on five representative human motion synthesis frameworks, Action2Motion Guo et al. (2020), ACTOR Petrovich et al. (2021), MotionDiffuse Zhang et al. (2024), MLD Chen et al. (2023) and state-of-the-art MoGenTS Yuan et al. (2024). More details can be found in section B of the appendix. We also report a comparison of model size and computational costs between our motion refinement method and the motion synthesis frameworks in section E.1 of the appendix.

### 4.2 Main Results

Experimental results before and after motion refinement on two datasets with five representative baselines are shown in Table 1. Results demonstrate that the proposed motion refinement method leveraging the motion concepts significantly rectifies the flawed frames and improves the synthesis quality in all the five evaluation metrics on average. Specifically, the improvements on FID and Accuracy are considerable, up to 25.00% and 3.68% respectively, demonstrating that our refinement framework can successfully bring realism and naturalness to synthesized human motions. Further,

| Methods | HumanAct12 / UESTC | | | | |
| --- | --- | --- | --- | --- | --- |
| | A-FID↓ | FID↓ | Accuracy↑ | Diversity→ | MultiModality→ |
| Real | 0.24 / 7.98 | 0.09 / 2.93 | 99.7 / 98.8 | 6.85 / 33.34 | 2.45 / 14.16 |
| Action2Motion | 5.80 / 164.27 | 2.46 / 53.34 | 92.3 / 83.2 | 7.03 / 26.31 | 2.87 / 13.99 |
| + Refined | 4.94 / 139.36 | 2.13 / 45.73 | 95.7 / 85.8 | 6.98 / 28.42 | 2.70 / 14.02 |
| *Impr.* | 14.83% / 15.16% | 13.41% / 14.27% | 3.68% / 3.13% | 27.78% / 30.01% | 40.48% / 17.65% |
| ACTOR | 1.24 / 89.03 | 0.12 / 20.49 | 95.5 / 91.1 | 6.84 / 31.96 | 2.53 / 14.66 |
| + Refined | 1.02 / 73.71 | 0.10 / 17.47 | 97.6 / 93.7 | 6.85 / 32.21 | 2.50 / 14.48 |
| *Impr.* | 17.74% / 17.21% | 16.67% / 14.74% | 2.20% / 2.85% | ∼100% / 18.12% | 37.50% / 36.00% |
| MotionDiffuse | 0.75 / 46.87 | 0.07 / 9.10 | 99.2 / 95.0 | 6.85 / 32.42 | 2.46 / 14.74 |
| + Refined | 0.63 / 38.07 | 0.06 / 7.52 | 99.4 / 96.7 | 6.85 / 32.68 | 2.45 / 14.58 |
| *Impr.* | 16.00% / 18.78% | 14.29% / 17.36% | 0.20% / 1.79% | 0.00% / 28.26% | ∼100% / 27.59% |
| MLD | 0.84 / 65.39 | 0.08 / 12.89 | 96.4 / 95.4 | 6.83 / 33.52 | 2.82 / 13.57 |
| + Refined | 0.65 / 52.34 | 0.06 / 9.93 | 98.6 / 97.0 | 6.84 / 33.48 | 2.71 / 13.69 |
| *Impr.* | 22.62% / 19.96% | 25.00% / 22.96% | 2.28% / 1.68% | 50.00% / 22.22% | 29.73% / 20.34% |
| MoGenTS | 0.65 / 42.15 | 0.06 / 8.56 | 99.4 / 96.2 | 6.85 / 33.47 | 2.46 / 13.72 |
| + Refined | 0.57 / 36.69 | 0.05 / 7.53 | 99.6 / 97.6 | 6.85 / 33.45 | 2.45 / 13.79 |
| *Impr.* | 12.31% / 12.95% | 16.67% / 12.03% | 0.20% / 1.46% | 0.00% / 15.38% | ∼100% / 15.91% |

Table 1: Performance comparisons of five baselines before and after the refinement across two benchmarks. All the baselines are implemented with their official codes EricGuo5513 (2021); Mathux (2022); mingyuan zhang (2023); ChenFengYe (2023); Yuan et al. (2024). Metric improvement calculations are detailed in section B.2 of the appendix.

| $\lambda$ | 100 | 200 | 300 | 400 |
| --- | --- | --- | --- | --- |
| MPJPE | 0.89 | 0.94 | 1.08 | 1.22 |
| Accuracy | 97.7% | **98.6%** | 98.3% | 97.1% |

Table 2: Comparison between different $\lambda$s in segmenting the motion sequence by dynamic programming on HumanAct12 Dataset with MLD. MPJPE reports the average fitting error of the twelve actions.

the diversity and multimodality of generated motions after refinement can be much closer to those of the real motions indicating the diversity and multimodality of motions can be greatly enhanced simultaneously.

As the motion concepts are learned action-by-action in our implementation, we provide the metric A-FID for a clearer and more precise vision of evaluating the quality of the learned concepts. The results show remarkable improvements, up to 22.62%, further exhibiting the effectiveness of these concepts. The per-action results can be found in section C of the appendix.

### 4.3 ABLATION STUDY

$\lambda$ **in Dynamic Programming**   We study the influence of $\lambda$ in segmenting the motion sequences using dynamic programming, shown in Table 2. The experiments are conducted on HumanAct12 with MLD as the baseline. We calculate the average fitting error represented in the form of MPJPE of the twelve actions in HumanAct12 and use the recognition accuracy to indicate the refinement performance. A larger $\lambda$ may reduce the number of the segmented transitions, which adds up to the fitting error at the same time. And if the $\lambda$ is too small, the anchors and transitions will be fragmentized and cannot well reveal the common and regular pattern of an action. We use $\lambda = 200$ for the dynamic programming process as a trade-off in our implementation.

**Clustering Algorithm**   We also conduct an ablation study on the algorithms for clustering the anchor data in Table 3. We use multiple widely-used clustering algorithms, namely K-means Likas et al. (2003), DBSCAN Ester et al. (1996), OPTICS Ankerst et al. (1999), and hierarchical clustering Ward Jr (1963) in our experiments. The results show that the refinement performance is insensitive to different clustering algorithms, indicating that the anchor concept can be learned stably. We use the K-means algorithm in our implementation as it demonstrated higher computational efficiency compared to other clustering algorithms.

More ablation studies can be found in section G of the appendix, including experiments on the contribution of anchor and transition refinement, comparison between spline representation and model fitting, comparison between our method and motion reconstruction, and analyses of the effects of $\lambda_{\mathrm{A}}$ and $\lambda_{\mathrm{KL}}$ in the loss of the CVAE transition generator.

| Algorithms | K-means | DBSCAN | OPTICS | Hierarchical |
|---|---|---|---|---|
| Accuracy | **98.6%** | 98.4% | 97.0% | 97.5% |

Table 3: Comparison among different clustering algorithms for anchor concept on HumanAct12 Dataset with MLD. The 'Hierarchical' refers to hierarchical clustering algorithm.

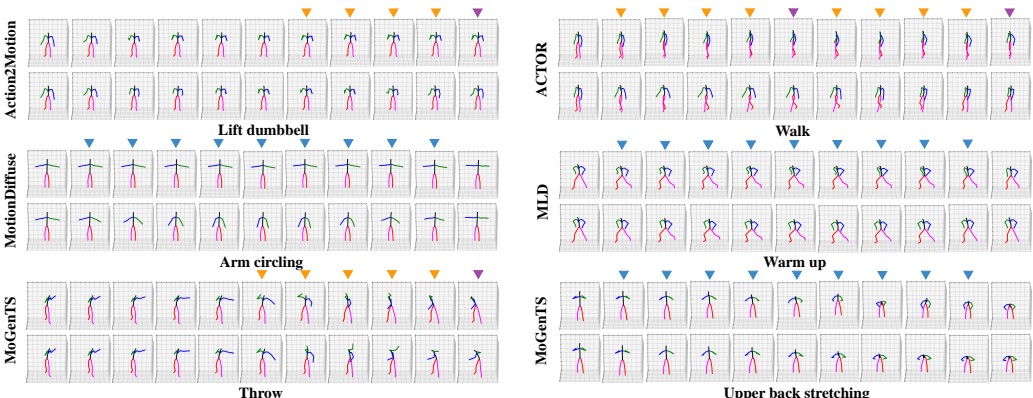

Figure 4: Visualizations of synthesized motion sequences before (top row) and after (bottom row) refining with our approach, across various baselines and actions. For a clear demonstration, we focus on visualizing the slice with flawed frames of the entire sequence, at regular intervals of 2 frames. Zoom in for a clear view.

## 4.4 QUALITATIVE ANALYSIS

We visualize the motion sequences before and after refinement across various baselines and actions in Figure 4 to comprehensively demonstrate how the flawed frames can be rectified with our approach. 'Action2Motion-LiftDumbbell', 'ACTOR-Walk' and 'MoGenTS-Throw' show common failure cases that occures in some flawed anchors. For 'Action2Motion-LiftDumbbell', the arms of the human should be put down eventually instead of being lifted up. The human in 'ACTOR-Walk' continuously takes steps with the same foot. For 'MoGenTS-Throw', the left arm is insufficiently extended, resulting in a less natural throwing posture. For these cases, our refinement framework can successfully correct the problems by using the right anchor and then providing proper transitions. 'MotionDiffuse-ArmCircling', 'MLD-WarmUp' and 'MoGenTS-UpperBackStretching' demonstrate three failure cases caused by flawed transition frames. For 'MotionDiffuse-ArmCircling', the arms are not put down during the motion. The leg of the human in 'MLD-WarmUp' is not pressed down, indicating a flawed transition. For 'MoGenTS-UpperBackStretching', the human temporarily returns to an upright posture during the forward-bending motion, rather than completing the movement smoothly. Our refinement framework successfully refines the flawed transitions with better realism. More qualitative analyses are provided in section H of the appendix.

## 5 CONCLUSION

In this paper, we study to solve the issue that flawed frames are inevitable to occur within a synthesized action-conditioned 3D human motion sequence. We propose motion concepts and correspondingly design a refinement network to improve the quality of a motion sequence. Motion concepts reveal the common and regular patterns in human actions, which can be unsupervisedly learned from a set of real motion sequences. We model the concepts from two aspects, the anchor concept and transition concept, to focus on the two typical cases in which flawed frames may occur. Taking advantage of these concepts, we design a refinement framework to recognize and refine the flawed frames in the given sequence with three modules, anchor recognition module, anchor refinement module, and transition refinement module. Experiments conducted on two widely used benchmarks with five representative motion synthesis approaches demonstrate that our refinement framework can effectively improve the realism of synthesized motions while simultaneously enhancing their diversity and multimodality.

## 6 ETHICS STATEMENT

Our work adheres to the ICLR Code of Ethics. This research focuses on refining action-conditioned synthesized motion sequences leveraging the learned motion concepts and does not involve human subjects, personally identifiable information, or sensitive data. All datasets used are publicly available and were collected and shared in accordance with their respective terms of use and licenses. We ensured that our models were trained and evaluated responsibly, and we followed standard practices for reproducibility and transparency. To help mitigate potential misuse of the proposed method, we will release our code and models under a license that discourages unethical use and include documentation outlining intended applications and limitations.

## 7 REPRODUCIBILITY STATEMENT

We have taken several steps to ensure the reproducibility of our results. The data collection process for learning the anchor and transition concepts is detailed in section 3.1.1 of the main text. All implementation details of our proposed method, including the model architecture, training settings, and hyperparameters, are provided in section 4.3 of the main text, and section A and section G of the appendix. A complete description of the evaluation metrics can be found in section B. Our source code along with a clear README file has been submitted as supplementary material for reproducibility.

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

## A  IMPLEMENTATION DETAILS

For our experiments, the root joint of the first frame of the input motion sequence is translated to the origin. The transformer in the motion encoder consists of 2 stacked transformer blocks with 4 attention heads and 128 FFN hidden dimensions. Both classifiers in our implementation are a three-layer MLP with an embedding dimension of 128. The decoder in the anchor refinement module is an MLP. The number of mixture components of the Gaussian Mixture Model (GMM) in the anchor recognition module is 2. In the transition refinement module, the bias encoder, the concept encoder, and the transformer have the same structure as the motion encoder, while the concept decoder is a three-layer MLP with an embedding dimension of 128. For training the anchor recognition module, anchor refinement module, and transition refinement module, we use the Adam optimizer with a base learning rate of 0.0005 and a polynomial decay with a power of 0.95. We set the batch size to 128 for all three modules with different epochs. The anchor recognition module is trained for 2000 epochs, while the anchor refinement module and the transition refinement module are for 5000 epochs. The training is carried out on a single RTX 2080Ti on Ubuntu 18.04 operating system. The total GPU memory usage is around 1,225MB.

## B  DATASETS, EVALUATION METRICS AND BASELINES

### B.1  DATASETS

Below are details of the datasets for action-conditioned motion synthesis:

- **HumanAct12 Guo et al. (2020)** is adapted from the PHSPD dataset Zou et al. (2020). The videos are temporally trimmed and only the joint coordinates in a canonical frame are provided. Specifically, all motions are organized into 12 categories and 34 subcategories with 1,191 motion sequences. We only use the coarse-grained action annotations of the 12 categories, following the common setting Guo et al. (2020); Petrovich et al. (2021). A body pose contains 24 joints.
- **UESTC Ji et al. (2018)** consists of 25K sequences organized into 40 action categories, most of which are exercises as well as some cyclic movements. Following Petrovich et al. (2021), we use the official cross-subject protocol to separate train and test splits, resulting in 10,650 sequences for training and 13,350 sequences for testing. A body pose contains 18 joints in the form of NTU-RGB-D dataset Liu et al. (2019).

B.2    EVALUATION METRICS

Below are details of the five metrics:

- **Frechet Inception Distance (FID).** FID serves as an important metric to evaluate the overall quality of generated motions. Features are extracted from a total of 3,000 generated motions and real motions for all action categories in the dataset. Then FID is calculated between the feature distributions of generated motions and the real motions. A lower FID suggests a better result.

- **Action Frechet Inception Distance (A-FID).** A-FID evaluates the FID of the generated motions within each action category. For each category, we sample 200 generated motions and real motions for FID calculation.

- **Recognition Accuracy.** The pre-trained action recognition model is used to classify the 3,000 generated motions and calculate the overall recognition accuracy.

- **Diversity.** Diversity measures the variance of the generated motions across all action categories. Two subsets of the same size $S_{\mathrm{d}}$ are randomly sampled from a set of generated motions of all action categories. These two sets of motion feature vectors are $v_1, ..., v_{S_{\mathrm{d}}}$ and $v'_1, ..., v'_{S_{\mathrm{d}}}$. The calculation of diversity is defined as

$$Diversity = \frac{1}{S_{\mathrm{d}}} \sum_{i=1}^{S_{\mathrm{d}}} ||v_i - v'_i||_2. \tag{6}$$

  $S_{\mathrm{d}} = 200$ is used in our experiments.

- **Multimodality.** Multimodality measures the diversity of the generated motions within each action category. For the $c$-th action category, we randomly sample two subsets of the same size $S_{\mathrm{m}}$ from a set of motions with $C$ action categories. These two sets of motion feature vectors are $v_{c,1}, ..., v_{c,S_{\mathrm{m}}}$ and $v'_{c,1}, ..., v'_{c,S_{\mathrm{m}}}$. The calculation of multimodality is defined as

$$Multimodality = \frac{1}{C \times S_{\mathrm{m}}} \sum_{c=1}^{C} \sum_{i=1}^{S_{\mathrm{m}}} ||v_{c,i} - v'_{c,i}||_2. \tag{7}$$

  $S_{\mathrm{m}} = 20$ is used in our experiments.

The calculation of these metrics relies on the motion feature extracted by a pre-trained action recognition model. For experiments on both datasets, we adopt the recognition models according to Petrovich et al. (2021). For experiments on HumanAct12, we directly use the provided recognition models of Guo et al. (2020) that operate on joint coordinates. For UESTC, we train a new joint-based recognition model with the same structure as described in Petrovich et al. (2021), as the pre-trained model in Petrovich et al. (2021) operates on pose parameters expressed as 6D rotations. The number of algorithm runs used to compute each reported result is 20. The improvement of A-FID, FID is calculated by $\frac{baseline\_value - refined\_value}{baseline\_value}$. The improvement of Accuracy is calculated by $\frac{refined\_value - baseline\_value}{baseline\_value}$. The improvement of Diversity and MultiModality is calculated by $\frac{|baseline\_value - real\_value| - |refined\_value - real\_value|}{|baseline\_value - real\_value|}$.

B.3    BASELINE MODELS

Below are the detailed introduction of the five baselines:

- **Action2Motion.** Action2Motion proposes a conditional temporal VAE equipped with a Lie algebra pose representation to synthesize natural and diverse human motions. It generates the present pose leveraging the posterior distribution learned from previous poses as a learned prior.

- **ACTOR.** ACTOR designs a transformer-based VAE to learn an action-aware latent representation for human motions. By sampling from the learned latent space, ACTOR can synthesize variable-length motion sequences conditioned on a categorical action.

| Methods | Metrics | - | warm up | walk | run | jump | drink | lift dumbbell |
|---|---|---|---|---|---|---|---|---|
| real | Accuracy | - | 100.0 | 100.0 | 100.0 | 100.0 | 98.5 | 100.0 |
| | MultiModality | - | 2.79 | 2.44 | 1.97 | 2.11 | 2.35 | 2.66 |
| Action2Motion | A-FID | baseline | 5.55 | 8.46 | 5.72 | 3.82 | 5.71 | 2.61 |
| | | + Refined | 4.87 | 7.91 | 5.18 | 3.28 | 4.46 | 2.02 |
| | | *Impr.* | 12.25% | 6.50% | 9.44% | 14.14% | 21.89% | 22.61% |
| | Accuracy | baseline | 86.5 | 87.0 | 94.4 | 94.7 | 85.1 | 96.9 |
| | | + Refined | 91.4 | 89.5 | 96.3 | 96.7 | 90.1 | 99.5 |
| | | *Impr.* | 5.66% | 2.87% | 2.01% | 2.11% | 5.88% | 2.68% |
| | MultiModality | baseline | 3.51 | 3.45 | 2.74 | 2.65 | 3.01 | 2.71 |
| | | + Refined | 3.28 | 3.30 | 2.48 | 2.54 | 2.58 | 2.62 |
| | | *Impr.* | 31.94% | 14.85% | 33.77% | 20.37% | 65.15% | 20.00% |
| ACTOR | A-FID | baseline | 1.47 | 1.53 | 1.88 | 2.36 | 1.06 | 0.47 |
| | | + Refined | 1.18 | 1.23 | 1.58 | 2.04 | 0.89 | 0.39 |
| | | *Impr.* | 19.73% | 19.61% | 15.96% | 13.56% | 16.04% | 17.02% |
| | Accuracy | baseline | 92.5 | 98.3 | 98.2 | 92.2 | 92.5 | 97.1 |
| | | + Refined | 94.8 | 99.2 | 99.1 | 94.2 | 94.9 | 98.8 |
| | | *Impr.* | 2.49% | 0.92% | 0.92% | 2.17% | 2.59% | 1.75% |
| | MultiModality | baseline | 3.26 | 2.54 | 2.17 | 3.03 | 2.78 | 3.00 |
| | | + Refined | 3.12 | 2.52 | 2.13 | 2.92 | 2.72 | 2.94 |
| | | *Impr.* | 29.79% | 20.00% | 20.00% | 11.96% | 13.95% | 17.65% |
| MotionDiffuse | A-FID | baseline | 0.90 | 1.07 | 1.13 | 1.43 | 0.65 | 0.29 |
| | | + Refined | 0.75 | 0.88 | 0.91 | 1.13 | 0.58 | 0.26 |
| | | *Impr.* | 14.06% | 17.76% | 19.47% | 20.98% | 10.77% | 10.34% |
| | Accuracy | baseline | 99.0 | 99.4 | 99.3 | 98.9 | 99.0 | 99.1 |
| | | + Refined | 99.3 | 99.5 | 99.5 | 99.2 | 99.2 | 99.2 |
| | | *Impr.* | 0.30% | 0.10% | 0.20% | 0.30% | 0.20% | 0.10% |
| | MultiModality | baseline | 2.91 | 2.48 | 2.02 | 2.34 | 2.37 | 2.74 |
| | | + Refined | 2.89 | 2.46 | 2.00 | 2.29 | 2.36 | 2.72 |
| | | *Impr.* | 16.67% | 50.00% | 40.00% | 21.74% | 50.00% | 25.00% |
| MLD | A-FID | baseline | 1.01 | 1.19 | 1.27 | 1.59 | 0.73 | 0.34 |
| | | + Refined | 0.76 | 0.89 | 0.94 | 1.16 | 0.59 | 0.28 |
| | | *Impr.* | 24.75% | 25.21% | 25.98% | 27.04% | 19.18% | 17.65% |
| | Accuracy | baseline | 93.6 | 98.7 | 98.7 | 93.1 | 93.5 | 98.1 |
| | | + Refined | 95.3 | 99.4 | 99.2 | 96.3 | 96.6 | 99.2 |
| | | *Impr.* | 1.82% | 0.71% | 0.51% | 3.44% | 3.32% | 1.12% |
| | MultiModality | baseline | 3.41 | 3.29 | 2.63 | 2.59 | 2.93 | 2.70 |
| | | + Refined | 3.22 | 3.04 | 2.45 | 2.46 | 2.76 | 2.68 |
| | | *Impr.* | 30.65% | 29.41% | 27.27% | 27.08% | 29.31% | 50.00% |
| MoGenTS | A-FID | baseline | 0.78 | 0.87 | 0.93 | 1.14 | 0.61 | 0.29 |
| | | + Refined | 0.69 | 0.71 | 0.79 | 0.95 | 0.53 | 0.26 |
| | | *Impr.* | 11.54% | 18.39% | 15.05% | 16.67% | 13.11% | 10.34% |
| | Accuracy | baseline | 99.2 | 99.5 | 99.5 | 99.3 | 99.2 | 99.3 |
| | | + Refined | 99.4 | 99.7 | 99.7 | 99.5 | 99.5 | 99.5 |
| | | *Impr.* | 0.20% | 0.20% | 0.20% | 0.20% | 0.30% | 0.20% |
| | MultiModality | baseline | 2.89 | 2.47 | 2.01 | 2.28 | 2.37 | 2.71 |
| | | + Refined | 2.87 | 2.45 | 1.99 | 2.25 | 2.36 | 2.69 |
| | | *Impr.* | 20.00% | 66.67% | 50.00% | 17.65% | 50.00% | 40.00% |

Table 4: Per-action results of HumanAct12 across five baselines.

- **MotionDiffuse.** MotionDiffuse utilizes diffusion model for motion generation. It takes advantages of Probabilitic Mapping, Realistic Synthesis and Multi-Level Manipulation to generate diverse and plausible motions.

- **MLD.** MLD designs a powerful VAE and arrives at a representative and low-dimensional latent code for motion sequences. It performs a diffusion process on the motion latent space to generate plausible human motion sequences conforming to the action classes with reduced computational overhead.

- **MoGenTS.** MoGenTS quantizes each individual joint to one vector, generating a spatial-temporal 2D token mask for motion quantization, and then introduces the temporal-spatial 2D masking and spatial-temporal 2D attention to leverage the spatial-temporal information between joints for motion generation. Since MoGenTS is originally designed for text-conditioned generation, we adapt it to the action-conditioned setting and perform motion refinement to the synthesized motion sequences with our approach.

| Methods | Metrics | - | sit | eat | t.s.w. | phone | boxing | throw |
|---|---|---|---|---|---|---|---|---|
| real | Accuracy | - | 100.0 | 100.0 | 100.0 | 98.9 | 99.8 | 100.0 |
| | Multimodality | - | 1.67 | 1.70 | 2.09 | 2.99 | 1.95 | 2.10 |
| Action2Motion | A-FID | baseline | 4.38 | 3.56 | 2.84 | 10.73 | 8.30 | 7.87 |
| | | + Refined | 4.16 | 3.32 | 2.50 | 8.82 | 6.45 | 6.36 |
| | | *Impr.* | 5.02% | 6.74% | 11.97% | 17.80% | 22.29% | 19.19% |
| | Accuracy | baseline | 93.5 | 92.3 | 98.9 | 74.1 | 77.7 | 79.4 |
| | | + Refined | 95.2 | 94.8 | 99.5 | 80.9 | 85.6 | 86.7 |
| | | *Impr.* | 1.82% | 2.71% | 0.61% | 9.18% | 10.17% | 9.19% |
| | MultiModality | baseline | 2.24 | 2.22 | 2.04 | 4.25 | 3.67 | 3.46 |
| | | + Refined | 2.12 | 2.14 | 2.05 | 4.06 | 3.30 | 3.15 |
| | | *Impr.* | 21.05% | 15.38% | 20.00% | 15.08% | 21.51% | 22.79% |
| ACTOR | A-FID | baseline | 1.32 | 0.48 | 0.87 | 2.03 | 0.55 | 0.80 |
| | | + Refined | 1.12 | 0.38 | 0.71 | 1.64 | 0.46 | 0.65 |
| | | *Impr.* | 15.15% | 20.83% | 18.39% | 19.21% | 16.36% | 18.75% |
| | Accuracy | baseline | 99.0 | 97.8 | 99.5 | 91.1 | 97.1 | 97.3 |
| | | + Refined | 99.6 | 98.6 | 99.7 | 93.2 | 98.7 | 98.9 |
| | | *Impr.* | 0.61% | 0.82% | 0.20% | 2.31% | 1.65% | 1.64% |
| | MultiModality | baseline | 1.39 | 1.71 | 1.90 | 3.65 | 2.20 | 2.50 |
| | | + Refined | 1.43 | 1.70 | 1.93 | 3.58 | 2.17 | 2.43 |
| | | *Impr.* | 14.29% | ~100.00% | 15.79% | 10.61% | 12.00% | 17.50% |
| MotionDiffuse | A-FID | baseline | 0.64 | 0.32 | 0.53 | 1.23 | 0.33 | 0.49 |
| | | + Refined | 0.55 | 0.28 | 0.46 | 1.03 | 0.30 | 0.44 |
| | | *Impr.* | 14.06% | 12.50% | 13.21% | 16.26% | 9.09% | 10.20% |
| | Accuracy | baseline | 99.4 | 99.2 | 99.7 | 98.8 | 98.9 | 99.1 |
| | | + Refined | 99.5 | 99.4 | 99.8 | 99.1 | 99.1 | 99.4 |
| | | *Impr.* | 0.10% | 0.20% | 0.10% | 0.30% | 0.20% | 0.30% |
| | MultiModality | baseline | 1.60 | 1.71 | 2.03 | 3.15 | 2.01 | 2.21 |
| | | + Refined | 1.62 | 1.70 | 2.05 | 3.11 | 1.99 | 2.18 |
| | | *Impr.* | 28.57% | ~100.00% | 33.33% | 25.00% | 33.33% | 27.27% |
| MLD | A-FID | baseline | 0.73 | 0.37 | 0.61 | 1.35 | 0.38 | 0.56 |
| | | + Refined | 0.58 | 0.31 | 0.49 | 1.03 | 0.32 | 0.48 |
| | | *Impr.* | 20.55% | 16.22% | 19.67% | 23.70% | 15.79% | 14.29% |
| | Accuracy | baseline | 99.2 | 98.4 | 99.6 | 93.2 | 98.2 | 98.3 |
| | | + Refined | 99.4 | 99.1 | 99.7 | 97.3 | 99.0 | 99.1 |
| | | *Impr.* | 0.20% | 0.71% | 0.10% | 4.40% | 0.81% | 0.81% |
| | MultiModality | baseline | 2.18 | 2.14 | 2.05 | 4.06 | 3.41 | 3.23 |
| | | + Refined | 2.03 | 2.02 | 2.07 | 3.71 | 2.94 | 2.90 |
| | | *Impr.* | 29.41% | 27.27% | 50.00% | 32.71% | 32.19% | 29.20% |
| MoGenTS | A-FID | baseline | 0.61 | 0.32 | 0.51 | 0.98 | 0.34 | 0.47 |
| | | + Refined | 0.54 | 0.29 | 0.46 | 0.85 | 0.31 | 0.40 |
| | | *Impr.* | 11.48% | 9.38% | 9.80% | 13.27% | 8.82% | 14.89% |
| | Accuracy | baseline | 99.5 | 99.4 | 99.7 | 99.1 | 99.2 | 99.3 |
| | | + Refined | 99.7 | 99.6 | 99.8 | 99.4 | 99.4 | 99.5 |
| | | *Impr.* | 0.20% | 0.20% | 0.10% | 0.30% | 0.20% | 0.20% |
| | MultiModality | baseline | 1.73 | 1.72 | 2.13 | 3.13 | 2.00 | 2.19 |
| | | + Refined | 1.71 | 1.71 | 2.11 | 3.10 | 1.98 | 2.16 |
| | | *Impr.* | 33.33% | 50.00% | 50.00% | 21.43% | 40.00% | 33.33% |

Table 5: Per-action results of HumanAct12 across five baselines. *t.s.w.* in this table refers to the action *turn steer wheel*.

## C  PER-ACTION RESULTS

To have a clearer view of how our refinement framework performs on each action, we present the per-action results of the twelve actions in HumanAct12 in Table 4 and Table 5 for all the five baselines. Here we report the metrics of A-FID, Accuracy, and MultiModality, considering FID and diversity is equal to A-FID and Multimodality when evaluating on a single action. As is shown, the synthesis quality is significantly improved according to the metrics. Specifically, the improvements of *walk*, *run* and *jump* for MLD in A-FID are over 25%. For MoGenTS, the improvements of *walk*, *run*, *drink*, *eat* and *turn steer wheel* reach 50.00%. The remarkable improvements in various metrics demonstrate the superiority of our proposed method in refining diverse action categories.

| Methods | HumanAct12 | | | | |
|---|---|---|---|---|---|
| | A-FID↓ | FID↓ | Accuracy↑ | Diversity→ | MultiModality→ |
| Real | $0.24^{\pm0.02}$ | $0.09^{\pm0.01}$ | $99.7^{\pm0.1}$ | $6.85^{\pm0.05}$ | $2.45^{\pm0.04}$ |
| Action2Motion | $5.80^{\pm0.26}$ | $2.46^{\pm0.08}$ | $92.3^{\pm0.2}$ | $7.03^{\pm0.04}$ | $2.87^{\pm0.04}$ |
| + Refined | $4.94^{\pm0.23}$ | $2.13^{\pm0.07}$ | $95.7^{\pm0.2}$ | $6.98^{\pm0.04}$ | $2.70^{\pm0.04}$ |
| ACTOR | $1.24^{\pm0.09}$ | $0.12^{\pm0.00}$ | $95.5^{\pm0.8}$ | $6.84^{\pm0.03}$ | $2.53^{\pm0.02}$ |
| + Refined | $1.02^{\pm0.07}$ | $0.10^{\pm0.00}$ | $97.6^{\pm0.7}$ | $6.85^{\pm0.03}$ | $2.50^{\pm0.02}$ |
| MotionDiffuse | $0.75^{\pm0.06}$ | $0.07^{\pm0.00}$ | $99.2^{\pm0.1}$ | $6.85^{\pm0.02}$ | $2.46^{\pm0.02}$ |
| + Refined | $0.63^{\pm0.04}$ | $0.06^{\pm0.00}$ | $99.4^{\pm0.1}$ | $6.85^{\pm0.01}$ | $2.45^{\pm0.01}$ |
| MLD | $0.84^{\pm0.07}$ | $0.08^{\pm0.00}$ | $96.4^{\pm0.2}$ | $6.83^{\pm0.05}$ | $2.82^{\pm0.04}$ |
| + Refined | $0.65^{\pm0.05}$ | $0.06^{\pm0.00}$ | $98.6^{\pm0.1}$ | $6.84^{\pm0.04}$ | $2.71^{\pm0.03}$ |
| MoGenTS | $0.65^{\pm0.04}$ | $0.06^{\pm0.00}$ | $99.4^{\pm0.1}$ | $6.85^{\pm0.02}$ | $2.46^{\pm0.01}$ |
| + Refined | $0.57^{\pm0.03}$ | $0.05^{\pm0.00}$ | $99.6^{\pm0.1}$ | $6.85^{\pm0.01}$ | $2.45^{\pm0.01}$ |
| Methods | UESTC | | | | |
| | A-FID↓ | FID↓ | Accuracy↑ | Diversity→ | MultiModality→ |
| Real | $7.98^{\pm0.68}$ | $2.93^{\pm0.26}$ | $98.8^{\pm0.1}$ | $33.34^{\pm0.32}$ | $14.16^{\pm0.06}$ |
| Action2Motion | $164.27^{\pm12.93}$ | $53.34^{\pm1.04}$ | $83.2^{\pm0.3}$ | $26.31^{\pm0.19}$ | $13.99^{\pm0.07}$ |
| + Refined | $139.36^{\pm10.69}$ | $45.73^{\pm0.89}$ | $85.8^{\pm0.3}$ | $28.42^{\pm0.17}$ | $14.02^{\pm0.06}$ |
| ACTOR | $89.03^{\pm8.48}$ | $20.49^{\pm2.31}$ | $91.1^{\pm0.3}$ | $31.96^{\pm0.36}$ | $14.66^{\pm0.03}$ |
| + Refined | $73.71^{\pm7.16}$ | $17.47^{\pm1.93}$ | $93.7^{\pm0.3}$ | $32.21^{\pm0.33}$ | $14.48^{\pm0.03}$ |
| MotionDiffuse | $46.87^{\pm3.15}$ | $9.10^{\pm0.44}$ | $95.0^{\pm0.0}$ | $32.42^{\pm0.21}$ | $14.74^{\pm0.07}$ |
| + Refined | $38.07^{\pm2.53}$ | $7.52^{\pm0.39}$ | $96.7^{\pm0.0}$ | $32.68^{\pm0.18}$ | $14.58^{\pm0.06}$ |
| MLD | $65.39^{\pm0.57}$ | $12.89^{\pm0.11}$ | $95.4^{\pm0.1}$ | $33.52^{\pm0.14}$ | $13.57^{\pm0.06}$ |
| + Refined | $52.34^{\pm0.43}$ | $9.93^{\pm0.08}$ | $97.0^{\pm0.1}$ | $33.48^{\pm0.12}$ | $13.69^{\pm0.05}$ |
| MoGenTS | $42.15^{\pm1.12}$ | $8.56^{\pm0.21}$ | $96.2^{\pm0.0}$ | $33.47^{\pm0.13}$ | $13.72^{\pm0.05}$ |
| + Refined | $36.69^{\pm0.95}$ | $7.53^{\pm0.18}$ | $97.6^{\pm0.0}$ | $33.45^{\pm0.11}$ | $13.79^{\pm0.04}$ |

Table 6: Performances and statistical intervals with 95% confidence of five baselines before and after the refinement across two benchmarks.

| | Action2Motion | ACTOR | MotionDiffuse | MLD | MoGenTS | Ours |
|---|---|---|---|---|---|---|
| Model size | 0.34M | 14.83M | 238.43M | 475.69M | 213.61M | 6.23M |
| Time/s | 1.047 | 0.025 | 6.133 | 0.553 | 1.614 | 0.017 |

Table 7: Model size and computation costs of our framework compared to the baselines.

## D  PERFORMANCES AND CONFIDENCE INTERVALS

In Table 6, we provide the performances and corresponding statistical intervals with 95% confidence of five baselines before and after the refinement across two benchmarks. The results show that the proposed motion refinement method can significantly improve the realism and naturalness of synthesized human motions, while enhancing the diversity and multimodality of generated motions simultaneously.

## E  DISCUSSION OF MOTION REFINEMENT FRAMEWORK

### E.1  MODEL SIZE AND COMPUTATIONAL COSTS

Although our proposed motion refinement framework consists of three modules, each module is in fact relatively small, ensuring high efficiency. The model size and computational costs of the proposed motion refinement framework, in comparison to the baseline motion synthesis frameworks, are detailed in Table 7. Despite comprising three distinct modules, our framework remains a relatively compact model size. Additionally, it maintains a low computational cost, thereby ensuring efficient performance.

| Methods | HumanAct12 | | | | |
|---|---|---|---|---|---|
| | A-FID↓ | FID↓ | Accuracy↑ | Diversity→ | MultiModality→ |
| Real | 0.24 | 0.09 | 99.7 | 6.85 | 2.45 |
| Action2Motion | 5.80 | 2.46 | 92.3 | 7.03 | 2.87 |
| + anchor | 5.23 | 2.26 | 94.8 | 7.01 | 2.78 |
| + anchor&transition | 4.94 | 2.13 | 95.7 | 6.98 | 2.70 |
| ACTOR | 1.24 | 0.12 | 95.5 | 6.84 | 2.53 |
| + anchor | 1.09 | 0.11 | 96.9 | 6.84 | 2.52 |
| + anchor&transition | 1.02 | 0.10 | 97.6 | 6.85 | 2.50 |
| MotionDiffuse | 0.75 | 0.07 | 99.2 | 6.85 | 2.46 |
| + anchor | 0.69 | 0.06 | 99.3 | 6.85 | 2.45 |
| + anchor&transition | 0.63 | 0.06 | 99.4 | 6.85 | 2.45 |
| MLD | 0.84 | 0.08 | 96.4 | 6.83 | 2.82 |
| + anchor | 0.70 | 0.07 | 97.8 | 6.84 | 2.74 |
| + anchor&transition | 0.65 | 0.06 | 98.6 | 6.84 | 2.71 |
| MoGenTS | 0.65 | 0.06 | 99.4 | 6.85 | 2.46 |
| + anchor | 0.60 | 0.05 | 99.5 | 6.85 | 2.45 |
| + anchor&transition | 0.57 | 0.05 | 99.6 | 6.85 | 2.45 |
| Methods | UESTC | | | | |
| | A-FID↓ | FID↓ | Accuracy↑ | Diversity→ | MultiModality→ |
| Real | 7.98 | 2.93 | 98.8 | 33.34 | 14.16 |
| Action2Motion | 164.27 | 53.34 | 83.2 | 26.31 | 13.99 |
| + anchor | 147.18 | 48.16 | 85.2 | 27.38 | 14.01 |
| + anchor&transition | 139.36 | 45.73 | 85.8 | 28.42 | 14.02 |
| ACTOR | 89.03 | 20.49 | 91.1 | 31.96 | 14.66 |
| + anchor | 78.50 | 18.74 | 92.8 | 32.12 | 14.52 |
| + anchor&transition | 73.71 | 17.47 | 93.7 | 32.21 | 14.48 |
| MotionDiffuse | 46.87 | 9.10 | 95.0 | 32.40 | 14.74 |
| + anchor | 40.89 | 8.04 | 96.2 | 32.57 | 14.64 |
| + anchor&transition | 38.07 | 7.52 | 96.7 | 32.68 | 14.58 |
| MLD | 65.39 | 12.89 | 95.4 | 33.52 | 13.57 |
| + anchor | 57.29 | 11.03 | 96.5 | 33.50 | 13.65 |
| + anchor&transition | 52.34 | 9.93 | 97.0 | 33.48 | 13.69 |
| MoGenTS | 42.15 | 8.56 | 96.2 | 33.47 | 13.72 |
| + anchor | 39.20 | 8.02 | 97.1 | 33.46 | 13.76 |
| + anchor&transition | 36.69 | 7.53 | 97.6 | 33.45 | 13.79 |

Table 8: Ablation study on the contribution of anchor and transition refinement.

### E.2 MODULAR OPTIMIZATION STRATEGY

As the three modules in the motion refinement framework each have distinct and clear goals, they are optimized separately with well-defined loss functions. In this manner, the whole framework will converge toward a global optimum when each module achieves its optimal performance.

## F DISCUSSION OF DATA COLLECTION

In this paper, we propose a heuristic algorithm based on dynamic programming to unsupervisedly capture anchor and transition concepts with spline, an analytical algebraic representation. Intuitively, key frame extraction Piperagkas et al. (2017); Li et al. (2017); Huang & Wang (2019); Yao (2022) is another possible way to separate key frames from a motion sequence and collect our training data. However, this approach actually does not fit in our framework, considering i) it typically relies on labeled data, which cannot be performed unsupervisedly and ii) It cannot ensure that the transitions between key frames fit well within the spline representations, which act as strong guidance during motion refinement.

| Methods | HumanAct12 | | | | |
|---------|-------|------|-----------|-----------|-----------------|
| | A-FID↓ | FID↓ | Accuracy↑ | Diversity→ | MultiModality→ |
| Real | 0.24 | 0.02 | 99.7 | 6.85 | 2.45 |
| MotionDiffuse | 0.75 | 0.07 | 99.2 | 6.85 | 2.46 |
| + Directly fitting | 0.66 | 0.06 | 99.3 | 6.85 | 2.45 |
| + Spline fitting | 0.63 | 0.06 | 99.4 | 6.85 | 2.45 |
| MLD | 0.84 | 0.08 | 96.4 | 6.83 | 2.82 |
| + Directly fitting | 0.69 | 0.07 | 97.7 | 6.84 | 2.74 |
| + Spline fitting | 0.65 | 0.06 | 98.6 | 6.84 | 2.71 |
| MoGenTS | 0.65 | 0.06 | 99.4 | 6.85 | 2.46 |
| + Directly fitting | 0.61 | 0.05 | 99.5 | 6.85 | 2.45 |
| + Spline fitting | 0.57 | 0.05 | 99.6 | 6.85 | 2.45 |

Table 9: Ablation study on spline representation and model fitting.

| Methods | HumanAct12 | | | | |
|---------|-------|------|-----------|-----------|-----------------|
| | A-FID↓ | FID↓ | Accuracy↑ | Diversity→ | MultiModality→ |
| Real | 0.24 | 0.02 | 99.7 | 6.85 | 2.45 |
| MotionDiffuse | 0.75 | 0.07 | 99.2 | 6.85 | 2.46 |
| + Motion reconstruction | 0.73 | 0.07 | 99.2 | 6.85 | 2.45 |
| + Ours | 0.63 | 0.06 | 99.4 | 6.85 | 2.45 |
| MLD | 0.84 | 0.08 | 96.4 | 6.83 | 2.82 |
| + Motion reconstruction | 0.85 | 0.08 | 96.4 | 6.83 | 2.81 |
| + Ours | 0.65 | 0.06 | 98.6 | 6.84 | 2.71 |
| MoGenTS | 0.65 | 0.06 | 99.4 | 6.85 | 2.46 |
| + Motion reconstruction | 0.63 | 0.06 | 99.4 | 6.85 | 2.46 |
| + Ours | 0.57 | 0.05 | 99.6 | 6.85 | 2.45 |

Table 10: Ablation study on three-step motion refinement framework and motion reconstruction.

# G ABLATION STUDY

**Anchor and Transition Refinement**    Our refinement framework progressively refines the anchor and transition frames in the given synthesized motion sequence. In Table 8, we report the improvement of only applying anchor refinement module and applying both anchor and transition refinement modules, to analyze the contribution of each module. The results demonstrate that the anchor refinement makes the primary contribution in the overall improvement and the transition refinement further elevates the performance to a higher level. This finding is consistent with the roles that an anchor or transition plays in a motion sequence, *i.e.* the anchor indicates the leading dynamics of an action while the transition reflects the core of detailed movement patterns.

**Spline Representation v.s. Model Fitting**    We think of a transition in real motion as two parts, i) transition concept for the core movement and ii) behavioral bias for the variance among the motion behaviors of different individuals. By utilizing spline representation, we can largely clear behavioral bias to capture clean, smooth and more essential transition concepts. However, transition concepts learned by model fitting may be interfered by behavioral bias and therefore be noisier. We conduct experiments to compare the two practices on MotionDiffuse, MLD, and MoGenTS. The results in Table 9 demonstrate the effectiveness of spline representation.

**Motion Concepts v.s. Reconstruction**    To further demonstrate the advantages of leveraging motion concepts in motion refinement, we compare with a reconstruction baseline approach that is entirely learned by model fitting. Particularly, the baseline approach uses a motion encoder (as described in Figure 3 of the main paper) to encode the full motion sequence and a transformer to decode it. The results, presented in Table 10, highlight that our motion refinement framework significantly enhances overall performance. In contrast, merely reconstructing the motion sequence not only fails to yield improvements but may even degrade performance. This limitation arises because straightforward reconstruction of the entire sequence does not effectively identify flawed frames

| $\lambda_{\mathrm{A}}$ | 0.2 | 0.5 | 1.0 | 2.0 |
|---|---|---|---|---|
| Accuracy | 86.1% | **98.6%** | 93.3% | 86.0% |
| $\lambda_{\mathrm{KL}}$ | $1e-3$ | $1e-4$ | $1e-5$ | $1e-6$ |
| Accuracy | 41.9% | **98.6%** | 85.4% | 51.3% |

Table 11: Comparison of recognition accuracy among different hyper-parameters for transition concept on HumanAct12 with MLD. The default values are $\lambda_{\mathrm{A}} = 0.5$ and $\lambda_{\mathrm{KL}} = 1e-4$.

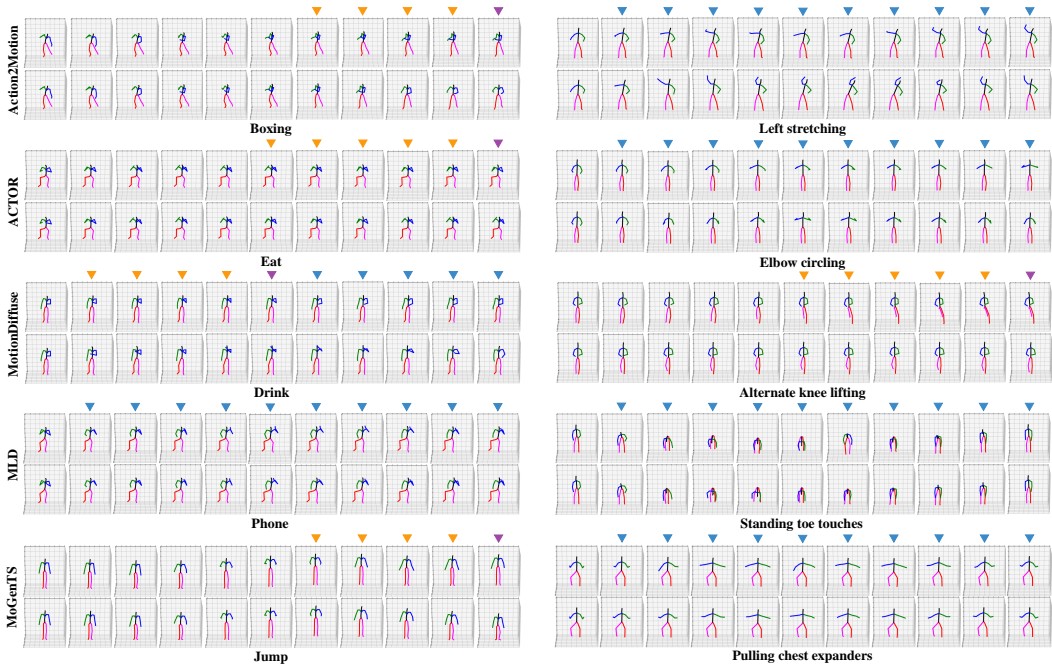

Figure 5: Visualizations of synthesized motion sequences before (top row) and after (bottom row) refining with our approach, across various baselines and actions. For a clear demonstration, we focus on visualizing the slice with flawed frames of the entire sequence, at regular intervals of 2 frames. Zoom in for a clear view.

or correct them accordingly. By leveraging the learned motion concepts, our framework addresses flawed anchor and transition frames, achieving superior generation results.

**$\lambda_{\mathrm{A}}$ and $\lambda_{\mathrm{KL}}$ in the Loss of the CVAE Transition Generator**   We study the influence of $\lambda_{\mathrm{A}}$ and $\lambda_{\mathrm{KL}}$ in the learning of transition concepts with CVAE. The experiments are conducted on HumanAct12 with MLD as the baseline. We use the recognition accuracy to indicate the refinement performance. As shown in Table 11, the learning of transition concepts is sensitive to both $\lambda_{\mathrm{A}}$ and $\lambda_{\mathrm{KL}}$. In our experiments, we set $\lambda_{\mathrm{A}} = 0.5$ and $\lambda_{\mathrm{KL}} = 1e-4$, as this combination yields the best refinement results.

## H   MORE QUALITATIVE RESULTS

To more comprehensively demonstrate the superiority of the proposed motion refinement method, we visualize more motion sequences before and after refinement across various baselines and actions in both HumanAct12 and UESTC benchmarks in Figure 5.

In the 'Action2Motion-Boxing' example, there is a failure case by synthesizing a flawed anchor at the end of the sequence, where the arms are held after punching. After refinement, the human puts his arms down to perform a complete boxing action. 'Action2Motion-Left-stretching' has a flawed transition that fails to fully press the body to the side to perform a perfect left stretching while only raising the right arm. Our refinement approach rectifies the flawed transition, leading

to better realism. 'ACTOR-Eat' is caused by a flawed anchor at the end of the sequence. After refinement, the human can eat continuously, which is more realistic. The elbows of the human in 'ACTOR-Elbow-circling' are not properly stretched, indicating a flawed transition. Through motion refinement, the movement of elbows becomes smooth and appropriate. In 'MotionDiffuse-Drink', there are flaws in both the anchor and transition: the arm is lifted too low to finish drinking, and the hand is put down too quickly. After refinement, the height of the arm is adequate for drinking, and the process of putting the hand down is smooth. In 'MotionDiffuse-Alternate-knee-lifting' example, the right knee is supposed to put down after lifting, while a flawed anchor results in the bending of the body. After refinement, the right knee is properly settled down along with a smooth transition. For 'MLD-Phone', the position of the phone is too distant for the human to hear within the transition. After refinement, the issue can be properly addressed. 'MLD-Standing-toe-touches' has flawed transitions that the body moves up and down repeatedly. Through motion refinement, the entire action is smooth with better realism. For 'MoGenTS-Jump', the anchor at the end of the sequence suggests the human remains suspended in the air, leading to less realistic motion. Our approach corrects this by generating a refined anchor in which the human has landed and is standing naturally. For 'MoGenTS-PullingChestExpanders', the right arm of the human is not fully raised to the side. Our approach can successfully correct this problem by generating a plausible transition that results in a proper arm position.

In conclusion, our approach can significantly enhance the visual quality with better realism by rectifying the flawed frames in both anchors and transitions.

# I    DISCUSSION ABOUT MOTION CONCEPTS AND MOTION PRIMITIVES

Our scope of application is very different from that of the motion primitives Kulal et al. (2022; 2021). In this paper, we focus on the task of refinement for **3D** synthesized motion sequences, while motion primitives are designed to construct motion programs capable of addressing **2D** video tasks. Therefore, the design of our approach is substantially different from the algorithms used in motion primitives. Specifically, we propose anchor and transition concepts that correspond to two aspects where flawed frames may occur, while motion primitives are designed as compact and human-interpretable representations for motion sequences and are not applicable for motion refinement.

To model these motion concepts, we apply anchor clustering to learn anchor concepts and utilize a CVAE-based transition generator conditioned on a pair of anchors to model transition concepts. This formulation better captures the correlation between anchor and transition concepts. In contrast, motion primitives model the motion sequence as a fixed number of consecutive primitives, with a separate distribution learned for the primitive at each position. This design does not explicitly account for the correlations between primitives across the sequence.

Building upon our proposed anchor and transition concepts, we design the three-step motion refinement framework to refine the flawed frames within 3D motion sequences hierarchically. The motion programs, composed of motion primitives, are used to address 2D video tasks, and cannot be applied to the refinement of 3D synthesized motions.

# J    LIMITATIONS, FUTURE WORK AND SOCIETAL IMPACTS

## J.1    LIMITATIONS

Since motion concepts are learned from real motion sequences, the refinement performance depends on the quality of the collected motion data. Limited training samples can lead to unstable concept learning and affect the refinement results. Since this paper focuses on motion refinement, we do not study the broader challenge of data scarcity in motion generation. We will address this challenge in future work.

## J.2    FUTURE WORK

As the refinement of synthesized motion sequences has rarely been studied before and it is extremely challenging and demands long-term research to develop a comprehensive refinement approach from the ground up, we begin our research from the task action-conditioned human motion synthesis.

For other tasks such as text-to-motion and audio-to-motion, our methodology has the potential to be adapted. We outline the pipeline for the text-to-motion task as an example. Given a text description and its corresponding synthesized motion sequence, we can first convert the text into an action sequence and segment the motion sequence accordingly. Each motion segment can then be refined using our motion refinement framework, which leverages the motion concepts learned for the corresponding action. Once all segments have been refined, the resulting motion sequence is expected to exhibit improved quality and realism. While this pipeline extends naturally from the methodology introduced in this paper, it presents several challenges, such as reliable action extraction from text and accurate motion segmentation based on the extracted action sequence. We aim to address these challenges in our future work and gradually extend our methodology to a broader range of human motion synthesis tasks.

Additionally, we hope that our work can draw interest within the community of motion refinement and facilitate further research on this problem.

### J.3 SOCIETAL IMPACTS

Our proposed motion concepts and correspondingly designed motion refinement framework can improve the realism of synthesized human motion sequences while simultaneously enhancing their diversity and multimodality. However, as motion refinement serves as a post-processing step for human motion synthesis, it requires extra computational resources to perform refinement, which could cost financial and environmental resources.

## K VIDEO

To better demonstrate the effectiveness of the proposed motion refinement method, we provide a video named *Cases_Before_and_After_Refinement.mp4* in the supplementary material to show the human motions before and after refinement of multiple actions from HumanAct12 and UESTC generated by various baselines.

## L CODES

We have submitted our code in the supplementary material. Please refer to *code/README.md* for more details.

## M THE USE OF LARGE LANGUAGE MODELS (LLMS)

Large Language Models (LLMs) do not play a significant role in our research. We only use LLMs to aid or polish writing.

