# OpenReview forum: "Refine Synthesized Action-Conditioned 3D Human Motion with Unsupervised Learned Motion Concepts"
_ICLR.cc/2026/Conference — ICLR 2026 Conference Withdrawn Submission_

### Official Review · Reviewer_bgTa · 2025-10-15

**Soundness:** 2
**Presentation:** 1
**Contribution:** 1
**Rating:** 2
**Confidence:** 5

**Summary:**

The paper proposes a refinement framework for synthesized 3D human motions. Instead of generating motion directly, it introduces unsupervisedly learned “motion concepts”—anchor and transition concepts—that capture high-level common patterns in motion sequences. Using these concepts, the framework refines flawed anchor frames and transitions in action-conditioned motion sequences through a three-step process: anchor recognition, anchor refinement, and transition refinement. Experiments on HumanAct12 and UESTC show improvements in FID, recognition accuracy, diversity, and multimodality across several baseline motion synthesis methods.

**Strengths:**

1. Concept-based representation: The anchor/transition concepts provide a structured way to capture high-level patterns in motion, making the refinement process interpretable to some degree.

**Weaknesses:**

1. Coordinate-based motion representation: The framework refines joint coordinates directly, rather than working in a rotation- or parameter-based representation (e.g., SMPL). This choice limits expressiveness and may reduce physical plausibility. Qualitative results also make it difficult to see what is truly improved, since coordinate-based visualization lacks richness.

2. Limited test cases: Experiments are restricted to small-scale, relatively clean datasets (HumanAct12, UESTC). It remains unclear how the method performs in the wild—for example, refining noisy 3D poses extracted from unconstrained videos, which is a much more realistic and challenging setting.

3. Clarity of writing and figures: The paper’s writing is sometimes unclear, and several figures are difficult to interpret. This reduces readability and makes it harder to judge the actual qualitative improvements.

**Questions:**

Please refer to Weaknesses.

---

### Official Review · Reviewer_7Qp9 · 2025-10-28

**Soundness:** 3
**Presentation:** 3
**Contribution:** 2
**Rating:** 4
**Confidence:** 2

**Summary:**

This paper tackles refinement of synthesized, action-conditioned 3D human motion sequences by learning unsupervised motion concepts from real data and then using those concepts to detect and fix flawed frames in generated sequences. The core idea is to view motions as consecutive transitions separated by anchors (key poses), fit each transition with a cubic spline per joint/axis, and then (i) cluster anchors into anchor concepts and (ii) learn a CVAE transition generator that outputs spline parameters conditioned on the two boundary anchors and duration. At inference, a three-step pipeline recognizes anchors and flags low-quality ones, replaces flawed anchors using the learned concepts, and regenerates transitions to ensure smoothness and coherence. Experiments on HumanAct12 and UESTC show consistent improvements when the refiner is applied to several representative baselines.

**Strengths:**

1. Clear motivation & task framing. The paper isolates a realistic pain point—flawed anchor/transition frames in synthesized actions—and argues for a dedicated refinement stage, a topic under-explored in prior work. The problem and the “anchor vs. transition” decomposition are well-motivated with examples.
2. Modular, practical refinement pipeline. A transformer-based module detects anchors and assigns quality scores (low score ⇒ refine), followed by anchor replacement with the learned concept and transition regeneration for continuity—simple to plug behind various generators.
3. Empirical signal across baselines. The refiner improves multiple action-conditioned baselines on standard datasets (HumanAct12/UESTC), with qualitative and quantitative evidence reported. (Main text claims; tables/figures referenced throughout.)

**Weaknesses:**

1. Spline as motion representation. The approach assumes per-joint, per-axis cubic splines (found via DP segmentation) can faithfully represent motion transitions. Is this assumption mathematically justified or empirically validated at scale? How robust is the spline representation near contacts/impacts/high-frequency events (e.g., footstrikes, abrupt direction changes, kicks, landings)? What happens under under/over-segmentation or imperfect fits—does the representation suppress legitimate high-frequency cues?

2. Where do the gains come from — representation  or concepts? The spline prior already imposes strong smoothness/low-pass behavior that might remove bias and stabilize transitions. It remains unclear how much improvement comes from this representation alone versus the unsupervised “motion concepts”. Mechanistic evidence of how concepts guide refinement would be helpful.

3. Generalization. How does the refinement work on zero-shot to out-of-distribution motions, for example, text-to-motion outputs, longer sequences, unseen actions?

4. Evaluation metrics. Beyond FID/A-FID/Accuracy/Diversity, evaluate perceptual and physical plausibility: foot sliding, bone-length stability, velocity/acceleration/jerk stats, trajectory smoothness, and human preference. Relevant references:

[1] Pose-NDF: Modeling Human Pose Manifolds with Neural Distance Fields (ECCV 2022)

[2] What is the Best Automated Metric for Text-to-Motion Generation? (SIGGRAPH 2023)

[3] Aligning Human Motion Generation with Human Perceptions (ICLR 2025)

5. Current visuals in the supplementary videos seems to have shown quality gain that is not so noticeable. Could the authors provide more visuals?

**Questions:**

1. Overall the scope is too narrow.

• Problem. The focus on action-to-motion is inherently limited; demonstrating extension or zero-shot transfer to text-to-motion would make the paper more applicable.

• Datasets. Using only UESTC (2018) and HumanAct12 (2020) makes the evidence dated and capacity-limited, so the conclusions are not fully convincing. Incorporating larger, more recent datasets (e.g., AMASS, Motion-X [4][5], MotionLib [6]) would strengthen the case.

• Task. How controllable is the motion refinement model? Could current structure be extended towards controllable editing, which could be a more generalized task?

[4] Motion-X: A Large-scale 3D Expressive Whole-body Human Motion Dataset.

[5] Motion-X++: A Large-Scale Multimodal 3D Whole-body Human Motion Dataset

[6] Scaling Motion Generation Models with Million-Level Human Motions (ICML 2025).

2. Could adaptive spline representations be applied as a more generalizable motion representation? How do the authors view this direction?

I am open to hearing the authors' opinions and willng to raise my rating if my concerns are addressed.

---

### Official Review · Reviewer_QZCM · 2025-10-30

**Soundness:** 3
**Presentation:** 2
**Contribution:** 2
**Rating:** 4
**Confidence:** 3

**Summary:**

This paper presents a novel refinement framework for action-conditioned 3D human motion synthesis, addressing the inevitable flawed frames in generated sequences. The method unsupervisedly learns motion concepts, anchor concepts (key poses) and transition concepts (movement connections), through dynamic programming segmentation. A three-stage refinement pipeline is designed: flawed anchor recognition, anchor refinement, and transition refinement. Extensive experiments on HumanAct12 and UESTC datasets demonstrate significant improvements across five representative motion synthesis methods, achieving up to 25% FID improvement while maintaining diversity and multimodality.

**Strengths:**

1. This work is the first to systematically address the inevitable flawed frames in action-conditioned 3D human motion synthesis through a dedicated refinement framework, filling an important research gap in this specific area.
2. The innovative concept of "motion concepts" - learning anchor and transition concepts unsupervisedly from real data, combined with the three-stage refinement pipeline (recognition, anchor refinement, transition refinement), demonstrates elegant methodological design.
3. The proposed refinement framework serves as a plug-and-play post-processing module that doesn't require retraining the original generation models, offering high computational efficiency and strong practical value with good scalability.

**Weaknesses:**

1. The reliance on a "quality score" from a GMM to identify flawed anchors lacks a clear, quantifiable objective standard (e.g., a distance threshold from cluster centers). This makes the flaw identification process somewhat of a black box, and its criteria may not be sufficiently transparent or robust.
2. The paper mentions the model size and inference time of the framework itself, noting it is much smaller than the generative models. However, as a post-processing step, it adds incremental cost to the entire generation pipeline. Whether this overhead is acceptable for real-time applications requires further discussion.
3. The current method might be proficient at fixing local, minor flaws (e.g., unnatural poses, unsmooth transitions). However, for severe, global logical errors (e.g., completely losing the gait rhythm in a "walking" action), this local refinement strategy may be insufficient.

**Questions:**

1. How would your refinement framework perform if a test action pattern is completely absent from the training data (out-of-distribution action), or if the generated motion contains severe physical implausibilities (e.g., limb penetration)? What do you perceive as the boundary of your method's capability in handling such complex flaws?
2. The paper mentions that the framework itself has low computational cost. However, when attached as a post-processing module to existing, potentially expensive generative models (e.g., diffusion models), what is the total computational overhead (generation + refinement)? Would this still be feasible for real-time applications?
3. The entire method heavily relies on the initial dynamic programming segmentation to define anchors and transitions. If this segmentation is itself inaccurate (e.g., missing a key anchor), how does this error propagate and affect the subsequent concept learning and refinement? How robust is the framework to such initial errors?

---

### Official Review · Reviewer_h38j · 2025-10-31

**Soundness:** 2
**Presentation:** 2
**Contribution:** 2
**Rating:** 2
**Confidence:** 3

**Summary:**

This paper tackles the problem of refining flawed frames in action-conditioned 3D motion generation. It uses a spline-based concept representation to clusters motion into two types: anchor concepts (key action states) and transition concepts (motion between anchors). Building on these concepts, the authors propose a three-stage refinement framework that detects flawed anchors via a Gaussian mixture model, and refines both anchors and transitions w.r.t extracted concepts. Results show that when applied to multiple baseline action-based motion generators, the framework consistently improves evaluation metrics.

**Strengths:**

The paper clearly identifies the underexplored problem of motion refinement in motion synthesis. The chosen spline-based motion concept representation is a well-motivated formulation that is supported by theoretical grounding from prior work and ablation studies. Visualizations show that the unsupervised clustering seems to capture semantically meaningful pose patterns for anchor concepts. The paper includes extensive comparisons per action categories to demonstrate the method’s quality and consistency.

**Weaknesses:**

- The clarity of presentation could be improved.
    - The visualizations (e.g., Figures 1 and 4) are difficult to interpret, where the depicted motion flaws seem to be subtle. While the supplementary video provides better visualization, many of the refinement examples also appear minor and may not convincingly demonstrate meaningful “flaws.” For instance, variations such as how high a dumbbell is lifted or how long an arm is raised while drinking do not strongly affect perceived motion quality. Including more pronounced failure cases, such as physically infeasible poses or clear inconsistencies with the action label (similar to the provided example of missing circular arm motion), would strengthen the motivation.
    - Figure 3 is a bit dense. Simplifying the pipeline layout and using more concise captions would improve readability.
    -  Figure 2’s caption should explicitly state that clustering is performed for anchor concepts.
-  The motivation and scope of the paper could be more convincing. As noted above, many of the presented flaws are quite subtle, which may partly be due to the simplified action-conditioning setup. In more general text-to-motion scenarios, errors such as text–motion inconsistencies are more common, but these may not be well addressed by the proposed framework.
- The anchor refinement stage resembles motion in-betweening, while the transition refinement behaves similarly to smoothing. It would be helpful if authors discuss or compare with existing text-conditioned motion in-betweening or masked motion models (e.g., Conditional Motion In-betweening [Kim et al., 2022], Flexible Motion In-betweening with Diffusion Models [Cohan et al., 2024], MoMask [Guo et al., 2024]) to clarify how the proposed method differs and whether those alternatives could serve as valid baselines for refinement.
- The benefit of motion concepts might partially come from regularizing generated motions toward the training distribution of the refinement model itself, whereas some of the evaluated motion generators (MLD, MotionDiffuse, etc.) are trained on a wider range of motions and for a more general task of text-to-motion generation. This raises the question of whether the observed improvements reflect genuine correction or simply smoothing and bias toward anticipated motion distributions, leading to better performance on the benchmark datasets. It should also be noted that text-to-motion generators, such as MLD, MotionDiffuse, may expect longer/more detailed text descriptions, hence may result in more "flaws" under action-conditioned settings.

**Questions:**

- Are there ever cases where no anchor frame is found to be flawed? Would the transition refinement still be conducted in such cases? It would be great to also see a transition refinement only ablation to compare with the provided anchor refinement only and anchor+transition refinement settings.
- Does the refinement process ever introduce new artifacts (e.g., temporal discontinuities, oversmoothing, or unnatural pauses)? If so, how were such cases handled?
- How were the numbers of clusters for anchor and transition concepts determined? Were they fixed across action categories, or tuned per action?
- Visualizations of the transition concepts (t-sne and motions) would further strengthen the presentation.
- As mentioned in the weaknesses section, it is unclear whether the refinement is simply biasing more general text-to-motion models towards the action-conditioned benchmark data distributions. It may be beneficial to compare the performance on data distributions that were not used for training the refinement framework, e.g. BABEL [Punnakkal, 2021].
- The authors note that per-action concepts are more effective for the task setting. Could the authors discuss the limitations of applying their framework with general concepts instead of per-action concepts and vice versa?

---

### Note · Authors · 2025-11-14

I have read and agree with the venue's withdrawal policy on behalf of myself and my co-authors.